# Analysing Early Diagnosis Strategies for HIV Infection: A Retrospective Study of Missed Diagnostic Opportunities

**DOI:** 10.3390/healthcare12030361

**Published:** 2024-01-31

**Authors:** Víctor Giménez-Arufe, Sandra Rotea-Salvo, Alejandro Martínez-Pradeda, Álvaro Mena-de-Cea, Luis Margusino-Framiñán, Jorge Suanzes-Hernández, María Isabel Martín Herranz, Purificación Cid-Silva

**Affiliations:** 1Service of Pharmacy, Biomedical Research Institute of A Coruña (INIBIC), University Hospital of A Coruña (CHUAC), Sergas, University of A Coruña (UDC), 15006 A Coruña, Spain; victor.gimenez.arufe@sergas.es (V.G.-A.); sandra.rotea.salvo@sergas.es (S.R.-S.); alejandromartinezpradeda@gmail.com (A.M.-P.); luis.margusino.framinan@sergas.es (L.M.-F.); isabel.martin.herranz@sergas.es (M.I.M.H.); 2Division of Clinical Virology, Biomedical Research Institute of A Coruña (INIBIC), University Hospital of A Coruña (CHUAC), Sergas, University of A Coruña (UDC), 15006 A Coruña, Spain; alvaro.mena.de.cea@sergas.es; 3Service of Infectious Internal Medicine, Biomedical Research Institute of A Coruña (INIBIC), University Hospital of A Coruña (CHUAC), Sergas, University of A Coruña (UDC), 15006 A Coruña, Spain; 4Clinical Epidemiology and Biostatistics Unit, Biomedical Research Institute of A Coruña (INIBIC), University Hospital of A Coruña (CHUAC), Sergas, University of A Coruña (UDC), 15006 A Coruña, Spain; jorge.suanzes.hernandez@sergas.es

**Keywords:** HIV/AIDS, HIV infection diagnosis, health screening, missed opportunities, delayed diagnosis

## Abstract

Early diagnosis of a Human Immunodeficiency Virus (HIV)-infected person represents a cornerstone of HIV prevention, treatment, and care. Numerous publications have developed recommendations where HIV serology is indicated to reduce missed diagnostic opportunities (MDOs). This retrospective study analyses new HIV infection diagnoses and the relationship between late diagnosis (LD)/advanced HIV disease (AHD), baseline characteristics, and MDOs. Sociodemographic data and data related to contact with the health system in the 5 years before diagnosis were collected. Most of the 273 diagnoses were made in primary care (48.5%). Approximately 50.5% and 34.4% had LD and AHD criteria, respectively. Female sex was associated with a higher incidence of LD. Persons infected through the heterosexual route and those at an older age had a higher risk for LD and AHD. People with previous HIV serology presented a lower percentage of LD and AHD. In total, 10% of the health contact instances were classified as MDOs, mostly occurring in primary care. A significant increase in the median of MDOs was observed in patients with LD/AHD. Female sex and hepatitis C virus co-infection were associated with an increase in the number of MDOs. The high percentage of LD and AHD and the significant number of MDOs show that the current screening system should be improved.

## 1. Introduction

Early diagnosis and treatment of people living with Human Immunodeficiency Virus (HIV) is essential to prevent disease progression. Otherwise, it may increase morbidity and mortality, possible virus transmission, and delay immunity recovery [1,2]. In addition, late diagnosis (LD) (CD4 lymphocyte count at diagnosis < 350 cells/μL) [3] is associated with increased costs, especially those related to hospital care [4]. However, early diagnosis of HIV is difficult, as people may not have symptoms for years.

In 2019, 2698 new HIV cases were reported in Spain. This was the first year in the last 10 years in which the number of new HIV cases decreased from the usual 3000+ reported [5]. The percentage of advanced HIV disease (AHD) cases (CD4 lymphocyte count at diagnosis < 200 cells/μL) was 25.1% in that year, and LD cases were 45.9%, with fewer cases in men who have sex with men (MSM) than in heterosexuals and in intravenous drug users (IVDUs) [3,6]. The data available for 2020 and 2021 regarding new HIV cases in Spain (1925 and 2786 cases, respectively) are likely to be influenced by the evolution of the SARS-CoV-2 pandemic.

It has been estimated that 14% of HIV-infected people in Spain are undiagnosed [7], and between 54% and 65% of new infections are caused by people unaware of their HIV status [8].

Promoting early diagnosis is one of the priority strategies of HIV prevention and care programmes in developed countries [9]. Different national and international organizations stress that more efforts are needed to achieve this. In addition, simple and inexpensive HIV diagnostic tests are now available.

In 2014, the Spanish Ministry of Health published the “Guide of recommendations for the early diagnosis of HIV in healthcare settings”, which considered routine testing as a viable option [10]. The Spanish guideline aligns with the World Health Organization and with the European guidelines [11,12,13]. These guidelines emphasize the importance of reducing missed diagnostic opportunities (MDOs) and make recommendations about in which situations the HIV serology would be indicated. These cases include conditions or diseases with an HIV prevalence greater than 0.1%, which are cost-effective and have the potential to enable earlier HIV diagnosis [14].

As reported in numerous articles, HIV-positive people who have not yet been diagnosed consult for many episodes at different levels of the healthcare system before diagnosis, generating MDOs [15,16].

This study aims to analyse new HIV diagnoses in our health area over the last 7 years and the relationship between LD/AHD, baseline sociodemographic characteristics, and the MDOs detected in these patients.

## 2. Materials and Methods

This was a retrospective, observational, and descriptive study of patients ≥ 18 years of age diagnosed and treated for HIV infection from a health catchment area of 550,000 inhabitants between 1 January 2013 and 31 December 2019.

Patients who transferred to our health area but had already been diagnosed and those who did not sign the corresponding informed consent form were excluded from the study.

Sociodemographic data and data related to HIV diagnosis (date of diagnosis, the level of care requesting the diagnostic test, previous serological determinations, route of infection, and immunovirological status of the patient), as well as the presence of other sexually transmitted infections (STIs) and exitus at the end of the study (31 December 2021), were collected.

The patient data were extracted from the Electronic Health Record, where there is information about contact with the health system at the three levels of care (primary care, specialised care, and emergency department) and analytical and other tests carried out in our health area and other health areas of the Autonomous Community of Galicia, in the northwest of Spain.

All contact with the health system at different levels of care in 5 years before the diagnosis of HIV infection (health contact (HC)) was analysed. A single HC instance was considered if the patient consulted several times for the same reason in <2 weeks in primary care. The recorded HC instances were categorised according to whether they corresponded to MDOs. They were considered HIV MDOs if the reason for the consultation met the criteria for “offer of HIV testing” in the guidelines for recommendations for the early diagnosis of HIV in healthcare settings [10] and no HIV diagnostic test was requested.

The statistical analyses were carried out with SPSS 24.0, except for the trend analyses, performed with the Joinpoint Regression Program, version 5.0.2. Numerical variables were presented as median and interquartile range for discrete variables and mean and standard deviation for continuous variables, and categorical variables were presented as frequencies using percentages and confidence intervals.

HC instances and MDOs were classified, and MDO prevalence by the level of care and other sociodemographic and immunovirological variables was analysed using the chi-square or Fisher’s exact test and the non-parametric Mann–Whitney U test, Kruskal–Wallis, or Spearman’s correlation coefficient for categorical and continuous variables, respectively. Repeated measurements were compared using the Wilcoxon test. This study investigated the impact of various factors on LD and AHD, including sociodemographic variables, MDOs, and the number of HC instances. The relationship between HIV test presence before diagnosis, the number of HC instances, and the immunological status at diagnosis (LD, AHD) was assessed using multivariate logistic regression.

All detected MDOs were grouped into four clusters by diagnostic test indicator status (AIDS-defining illnesses; illnesses associated with undiagnosed HIV prevalence > 0.1%; other illnesses considered likely to have undiagnosed HIV prevalence > 0.1%; and targeted offer (at-risk population)) [10,13], and their prevalences were analysed.

Finally, the effect of the study variables (odd ratio (OR) estimation) was determined using logistic regression models on the risk of LD or AHD (multivariate regression).

The *p*-values ≤ 0.05 were considered statistically significant.

This study was approved by the corresponding Research Ethics Committee (CEI A Coruña-Ferrol, 2014/564, date of approval 24 November 2014). This study was conducted in accordance with the Helsinki Declaration of Good Clinical Practice.

## 3. Results

A total of 273 patients diagnosed and treated for HIV infection were included. The mean age at diagnosis was 39.6 ± 11.0 years, with 61.2% of the patients in the 31–50 age group. Of the newly diagnosed patients, 86.1% were men and most were Caucasian (76.9%), although there was a high percentage of Latinos (people from or with origins that trace back to Latin America, including Central America, South America, and the Caribbean) (19.8%). The main mode of HIV transmission was sexual (86.1%), mainly MSM (57.9%). Viral hepatitis B or C co-infection was present in 7.7% of patients (Table 1).

At diagnosis, the median CD4 cell count was 345.0 (123–510) cells/μL, and the mean viral load was 5.09 ± 0.89 log copies/mL. Clinical classification is detailed in Table 1. AIDS criteria were present in 35.5% of the patients at diagnosis. Of the study population, 50.5% and 34.4% had LD and AHD criteria, respectively. No differences were observed when analysing the evolution of LD and AHD cases over the study period, and the trend remained stable (Table 2).

At the end of this study, five patients died, and only one death was related to HIV infection (pneumonia).

Most new diagnoses were made in primary care (48.5%) and in specialised care (38.1%). Of the patients diagnosed in specialised care, 17.6% were diagnosed during hospitalisation.

Regarding the influence of sociodemographic variables on LD and AHD (Table 3), women presented a higher number of LD and AHD compared with men, although it was only significant in the case of LD (*p* = 0.043). Furthermore, the mean age of patients with LD and AHD was higher than patients without LD and AHD (*p* < 0.001). Regarding race, the percentage of patients with AHD at diagnosis was higher in the Latino group compared with the Black group (*p* = 0.02).

A higher viral load at diagnosis was observed in those patients with LD and AHD (*p* < 0.001), and hepatitis C virus (HCV) co-infection was significantly related to AHD at diagnosis (*p* = 0.032).

A lower percentage of LD and AHD at diagnosis was found among MSM patients compared to the other modes of HIV transmission.

The number of patients with LD is independent of the level of care. However, a higher percentage of patients with AHD was observed among those diagnosed in an inpatient episode than those diagnosed in primary care (*p* = 0.004).

In total, 30.4% of patients had at least one negative HIV serology 5 years before diagnosis. The population with prior serology had a significantly lower percentage of LD, AHD, and Acquired Immunodeficiency Syndrome (AIDS) than those without (Table 4).

In the logistic regression analysis, the variables implicated in an increased risk of LD were age (each year increase in age increases the risk of AHD by 3%), viral load, and route of transmission (heterosexual vs. MSM multiplies the risk by 2.2 and unknown route multiplies the risk by 4.1) (Table 5).

The variables that increased the risk of AHD were age (each year increase in age increases the risk of AHD by 3%), transmission route (heterosexual vs. MSM increases the risk of AHD by 2.3 times compared to MSM, and unknown route increases the risk by 4.7 times), and HCV co-infection (HCV co-infection increases the risk of AHD by 10 times) (Table 5).

In the study population, a total of 1,987 HC instances were analysed. Of these consultations, 63.7% were performed in primary care, 21.7% in the emergency department and, 14.2% in specialised care consultations. The number of HC instances performed by level of care, as well as the mean per patient, is detailed in Table 6.

The relationship between the number of HC instances, LD, and AHD was analysed. Patients who met AHD criteria at diagnosis had more HC instances (*p* = 0.039). This association was not detected in the case of LD (*p* = 0.386).

Of all the HC instances analysed, 233 were considered MDOs (11.7%). A total of 57.9% of the MDOs occurred in primary care, while 23.6% and 18.5% were found in emergency department and specialised care consultations, respectively. When analysing the MDOs per patient, one MDO was detected in 83 patients, two to three MDOs in 36 patients, and only 2 patients had five and six MDOs. No MDOs were detected in the remaining patients diagnosed in the study period (152; 55.7%).

An average annual decrease of 12.39% (24.39–1.53%) was observed in the number of MDOs detected yearly, which was significant despite the increase in 2018. The trend remained constant over the entire period (Figure 1).

The five most frequent MDOs were STIs (42), candidiasis (28), seborrhoeic dermatitis/exanthema (24), unjustified weight loss (19), and pneumonia (17). Breaking down the MDOs by the level of care, the two most frequent MDOs in the emergency department were pneumonia (12) and fever without apparent cause (9); in specialised care outpatient clinics, STIs (12) and seborrhoeic dermatitis/exanthema (9); and in primary care, STIs (27) and seborrhoeic dermatitis/exanthema (14) (Table 6). The MDOs detected were grouped into four groups by indicator condition, and their prevalence was analysed, which is detailed in Figure 2.

The only baseline characteristics associated with an increased risk of MDOs were sex (higher number of MDOs in women) (*p* = 0.012) and HCV co-infection (*p* = 0.032).

A significant increase in median MDOs was observed in patients with LD (*p* < 0.001) or AHD (*p* < 0.001) at diagnosis (Table 6).

## 4. Discussion

Our study showed that half of the new HIV diagnoses in our health area were LDs, and one-third were AHD cases. Women had a higher incidence of LD, and older people were more at risk of having LD or AHD. Also, heterosexual patients had a higher risk of LD or AHD, in contrast to MSM. Contrary to expectations, almost 12% of HC instances were considered MDOs and were more frequent in women and HCV-co-infected individuals. Probably most importantly, patients with LD or AHD had more MDOs, a critical point to try to improve HIV screening.

The baseline characteristics of new HIV diagnoses in our health area resembled those described in other studies [17] and the published national data [6], with a similar mean age at diagnosis (39.6 vs. 36.0), a higher prevalence of new diagnoses among the population aged 31–50 years, a predominance of new diagnoses among men (86.1% vs. 84.3%), and sexual intercourse as the main factor of transmission (86.1% vs. 82.7%), predominating in MSM (57.9% vs. 55.2%). Our findings align with previous research, indicating that the patient profile has remained consistent in recent years. As in our study, other authors, such as Mínguez-Gallego et al., observed an increase in foreign patients among the new diagnoses, with the population of Latino origin standing out (19.8% vs. 22.4%) [6]. It might be necessary to update the current guide, as there has been a shift in the profile of the diagnosed population. Conducting a serology test for this population during their first interaction with the healthcare system could be beneficial. Most new diagnoses were detected in primary care (48.5%). This contrasts with other populations, such as the USA, where most diagnoses occur in infectious disease clinics or during hospital admission [18,19,20]. However, we observed similar results in populations with similar healthcare systems [21,22], where the cornerstone of the system is in primary care. In our study, the low percentage of patients diagnosed in hospitalisation processes (17.6%) is noteworthy when compared with reports of other studies [20]. However, Mínguez-Gallego et al., in their study, detected an upward trend in diagnoses during hospitalisation [17]. On the other hand, Levy et al. highlight how AHD patients accounted for more than 50% of inpatient diagnoses [23]. It is important to note that the emergency department did not diagnose any patients with HIV. Serology tests were only conducted on those who were hospitalised or referred to specialist care consultations. This could be because serology tests are voluntary and require informed consent from the person being tested, along with a brief pre-test information session, as per our country’s regulations. Furthermore, the limited time available to care for patients in emergency departments may also contribute to the lack of HIV diagnoses.

One effective approach to improve diagnoses in emergency departments is to emphasize and disseminate guidelines for normalizing their use and reducing the number of procedures needed to request serology.

The results of the median CD4 counts obtained in our study are comparable to those reported in the epidemiological surveillance report on HIV and AIDS in Spain in 2020 (345.0 vs. 371.0) [6]. Similar data were reported in other MDO studies, such as those of Gullón et al. and Hopkins et al. (352.0 and 404.0, respectively) [24,25]. Similar to those of most national studies [24,26], CD4 counts differed according to the transmission route, being lower among MSM patients with LD and AHD. In our study, we observed an increase in LD and AHD with increasing patient age and HIV viral load at diagnosis, as well as a higher number of LD in women. According to the available data, the percentage of new diagnoses in the female sex decreases yearly in Spain [6]. However, according to national data, they are more likely to have LD, confirmed in our population. Possible explanations for this finding could be related to low risk awareness or lack of disease awareness compared to more exposed populations. Furthermore, our results showed a greater number of AHD in HCV-co-infected patients, similar to that reported by Gullón et al. [24]. The poor immunological condition of these patients may be a result of comorbidities and the progression of the disease worsened by this factor. However, it could also indicate a failure to follow the recommended guidelines, especially since most of the co-infected patients are IVDUs.

Despite the recommended early diagnostic measures, the percentage of patients with LD and AHD at the time of diagnosis remains high. Gargallo-Bernad et al. and Gullón et al. have published LD percentages similar to our population [21,24], although the national average has been lower [27], and neighbouring countries have reported higher values [28]. We observed a higher percentage of patients with AHD or AIDS at the time of diagnosis than in other national and international studies [24,27,28]. However, there are studies in other regions of Spain with similar results to ours [29]. In contrast to other North American studies [30], in our setting, no association of LDs with the place of diagnosis was seen. However, an increase in AHD was observed in patients diagnosed with HIV during hospitalisation, which is logical, as patients are usually admitted for advanced clinical symptoms of immunosuppression. This again underscores the importance of primary care diagnosis in reducing AHD incidence [31].

The presence of prior serology significantly decreased the number of patients with LD, AHD, and AIDS. These results agreed with those obtained in other studies [26,29]. Both studies were carried out in similar health systems, where free access to this type of test is guaranteed. This ease of access may be the key factor that explains the relationship between the presence of a previous serology and the lower number of LD, AHD, and AIDS cases. This highlights the importance of establishing tools for more agile and population-wide screening. An example that is easy to implement and is currently being used to achieve HCV “microelimination” is to perform serology in the entire population in a specific age range and make a consultation in primary care, regardless of the reason for the consultation [32,33]. Despite early diagnosis guidelines, improved diagnostic circuits, and new tools available (rapid tests and others), the percentage of patients with LD and AHD remained constant throughout our study period. These results are in agreement with those reported nationally [6]. According to WHO guidelines, it is important to tailor guidelines based on the specific characteristics of the target population. An update to the guidelines may be necessary based on changes in the diagnostic profile, lack of screening identified in emergency departments, and stagnation in improving CD4 counts.

Although there was a diversity of results, the high percentage of LDs and ADs showed that the current diagnostic method [10], mainly based on testing patients with pathologies most likely linked to HIV infection, is not effective enough. It relies primarily on the person having a health contact instance, and secondly, the physician must recognise the current pathology as a signal for HIV testing. Neilan et al. conclude that routine screening at age 25 is more cost-effective than the usual recommendations [34]. Given the costs of current tests, establishing age ranges among the sexually active and carrying out systematic screening can be an interesting tool to reduce the number of late diagnoses. Another tool implemented with promising results is automating diagnostic screening recommendations using algorithms in the Electronic Health Record [35]. The automation of these systems, feasible in any health system with electronic medical records, can contribute to reducing the stigma associated with these tests and normalizing them, reducing the time until diagnosis. The emergence of rapid tests and self-testing may explain why the percentage of patients with previous serology was similar to other national and international articles (30.4% vs. 29.9% vs. 27.3% vs. 27.4%) [25,26,29] and lower than reported before the advent of these tests [36]. However, standardisation of these tests remains essential. According to the CDC (Centers for Disease Control and Prevention), patients consult an average of 2.5 times in primary care before a diagnostic test is ordered [37]. Rapid tests could decrease the time to diagnosis, but there is a need to raise awareness in society for more widespread use.

As in our population, patient age at diagnosis and mode of transmission are common risk factors for LD and AHD in most studies [17,21,24,38]. Furthermore, we observed in our population that HCV co-infection increased the likelihood of being diagnosed with a CD4 count < 200 cells/μL by 10-fold. As mentioned before, this might be attributed to the poor immunological condition of these patients. The results of national studies such as the one by Mínguez-Gallego et al. [17], which are in line with our findings, show that geographical origin is not found to be a risk factor for LD, AHD, or both, which could be due to the universal access to healthcare in our country. This is in contrast to studies conducted outside Spain [39]. However, Gullón et al. and Gargallo-Bernad et al. also found an association between LD and the immigrant population in Spain [21,24]. These differences within the same healthcare system may be attributed to the regions where the study was conducted, with a larger impact in areas where there is a higher number of immigrants.

Comparing the identified HC instances in this study with those of other studies is difficult, as there are many ways to classify them (including only HC instances produced in the emergency department, in a single centre, etc.), or they directly reflect the number of MDOs without indicating the analysed HC instances [21,24,30]. Most of the HC instances in our study came from primary care, which is expected considering the structure of our health system [21]. When studying the relationship between the number of HC instances with LD and AHD, we detected that the population with a higher number of HC instances had a lower CD4 count at the time of diagnosis, in line with other published studies [16,19] and reaching statistical significance in AHD. Of all the analysed HC instances, 11.7% were classified according to established criteria as MDOs. Powell et al. state that 25% of new diagnoses with previous HC instances had at least one indicator condition for HIV serology [16]. When breaking down MDOs by healthcare level at diagnosis, we found that the majority occur in primary care, as was the case for the number of diagnoses and HC instances. Studies with similar health systems show identical results [21,40]. Overall, the number of MDOs remained stable with slight fluctuations over the years, highlighting the few improvements that have been made in the screening process. As in most published studies [21,23,24,40,41], the most frequent MDOs are STIs (18.0%), followed by candidiasis and seborrhoeic dermatitis/exanthema. When MDOs were broken down by healthcare level, STIs were the main cause of MDOs in primary care and specialised care. However, in the emergency department, the main cause was pneumonia, identified by both Powell et al. and Nanditha et al. as the leading cause of MDOs [16,42]. The high percentage of MDOs in patients consulting for STIs again highlights how current screening systems are not fully effective, as the diagnosis of an STI does not determine HIV screening. The opposite is true for diagnosing Pneumocystis jirovecii infection, where no MDOs were detected.

The main limitations of our study were the small sample size compared with other populations studied and the single-centre retrospective observational design, which could have introduced uncontrolled bias. In addition, when extracting information from computerised medical records, information that was not transcribed or not correctly coded (reason for consultation, diagnosis, and others) could have been lost. All contact instances analysed were captured from a single public centre; therefore, any other contact in other private or public centres may lead to underestimating the MDOs. The articles reviewed showed high variability in study populations and ease of access to diagnostic tests across different health systems. In addition, there was no consensus on the definition of LD and AHD, further complicating the comparison of different studies. There were also differences in the classification of indicator conditions considered MDOs, making comparing different studies difficult. Furthermore, the fact that the WHO guidelines suggest that each country must adapt its guidelines to the target population makes the comparison between studies more difficult.

Further studies comparing the current HIV serology system by indicator pathology with screening algorithms based on information available in the medical record or with population-based screening based on rapid tests would be desirable.

## 5. Conclusions

Despite recommendations to perform an HIV diagnostic test upon detection of one of the indicator conditions, this study shows that the number of MDOs remains considerable, resulting in a higher percentage of patients with LD and AHD among the population with detected MDOs. The number of LDs and AHD decreases among the population with previous serology. Implementing other strategies for better population screening is essential in areas such as emergency departments and especially among subgroups with a higher risk of experiencing MDOs and being diagnosed in more advanced stages of immunosuppression.

## Figures and Tables

**Figure 1 healthcare-12-00361-f001:**
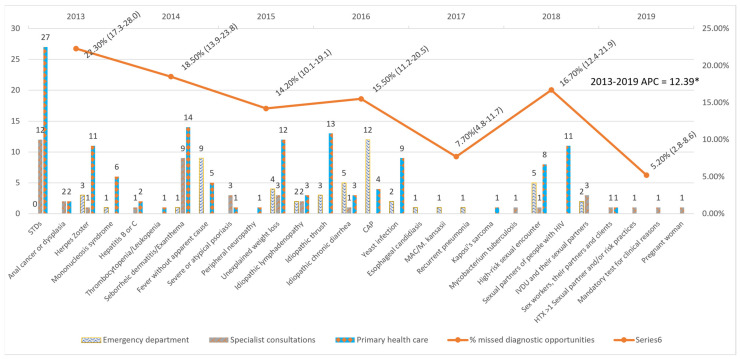
Missed diagnostic opportunities by indicator condition and evolution of the percentage of Missed diagnostic opportunities per year. STDs: Sexually transmitted diseases; CAP: Community-acquired pneumonia; MAC: Mycobacterium avium complex; IVDU: Intravenous drug users; HTX: heterosexual; APC: Annual Percent Change. * APC is significantly different from zero at the alfa = 0.05 level.

**Figure 2 healthcare-12-00361-f002:**
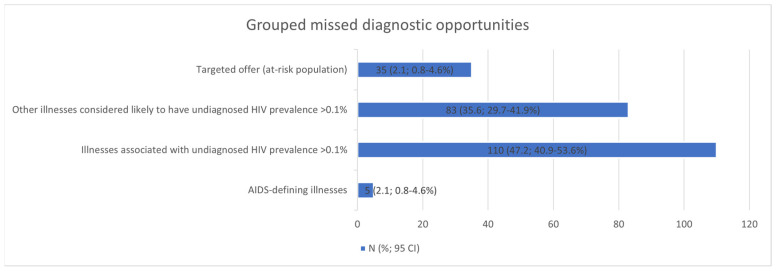
Missed diagnostic opportunities grouped into four clusters. CI: confidence interval.

**Table 1 healthcare-12-00361-t001:** Sociodemographic and immunovirological variables and level of diagnostic care.

Total Population (N = 273)					
Sex		N (%)		CI 95%	
Male		235 (86.1)		81.5–89.7	
Female		38 (13.9)		10.3–18.5	
Age (years)		Mean		SD	
		39.6		11.0	
Age (by groups)		N (%)		CI 95%	
≤30		61 (22.3)		17.8–27.6	
31–40		78 (28.6)		23.5–34.2	
41–50		89 (32.6)		27.3–38.4	
51–60		35 (12.8)		9.4–17.3	
61–70		9 (3.3)		1.7–6.1	
>70		1 (0.4)		0.1–2.0	
Race		N (%)		CI 95%	
Caucasian		210 (76.9)		71.5–81.5	
Latino		54 (19.8)		15.5–24.9	
Black		9 (3.3)		1.7–6.1	
Co-infection		N (%)		CI 95%	
Hepatitis B virus		6 (2.2)		1.0–4.7	
Hepatitis C virus		15 (5.5)		3.4–8.9	
HIV transmission mode		N (%)		CI 95%	
MSM		158 (57.9)		51.9–63.6	
Heterosexual		77 (28.2)		23.2–33.8	
IVDU		15 (5.5)		3.4–8.9	
Unknown		23 (8.4)		5.7–12.3	
Patients with previous serology		N (%)		CI 95%	
		83 (30.4)		25.2–36.1	
Immunological status		Mean	Median	SD	IQR
CD4 (cells/μL)		--	345.0	--	123–510
Viral load (log copies/mL)		5.09	--	0.89	--
AIDS		N (%)		CI 95%	
		97 (35.5)		30.1–41.4	
Late diagnosis (<350 cells/μL)		N (%)		CI 95%	
		138 (50.5)		44.7–56.4	
Advanced HIV disease (<200 cells/μL)		N (%)		CI 95%	
		94 (34.4)		29.0–40.2	
Exitus		N (%)		CI 95%	
		5 (1.8)		0.8–4.2	
Level of diagnostic care		N (%)		CI 95%	
Primary care		134 (49.1)		43.2–55.0	
Specialised care consultation		56 (20.5)		16.1–25.7	
Inpatient		48 (17.6)		13.5–22.5	
Private consultation		25 (9.1)		6.3–13.2	
NGO/“ADOS”		4 (1.5)		0.6–3.7	
Rapid test		6 (2.2)		1.0–4.7	
Clinical category N (%)					
	1	2		3	
A	69 (25.3)	96 (35.2)		43 (15.7)	
B	2 (0.7)	9 (3.3)		15 (5.5)	
C	1 (0.4)	3 (1.1)		35 (12.8)	

CI: confidence interval; SD: standard deviation; IQR: interquartile range; MSM: men who have sex with men; IVDU: intravenous drug users; AIDS: Acquired Immunodeficiency Syndrome; NGO: non-governmental organization; ADOS: axencia de doazón de órganos e sangue (Galician blood and tissue donation agency). Clinical category: A: asymptomatic; B: not A or C category; C: AIDS indicator condition; 1: ≥500 cells/μL; 2: 200–499 cells/μL; 3: <200 cells/μL.

**Table 2 healthcare-12-00361-t002:** Late diagnosis/advanced HIV disease time evolution.

Year	Total Diagnoses (N)	Late Diagnosis (N (%))	Advanced HIV Disease (N (%))
2013	58	31 (53.4)	21 (36.2)
2014	48	24 (50.0)	18 (37.5)
2015	44	19 (43.2)	12 (27.3)
2016	46	23 (50.0)	14 (30.4)
2017	21	11 (52.4)	9 (42.9)
2018	31	20 (64.5)	14 (45.2)
2019	25	10 (40.0)	6 (24.0)

**Table 3 healthcare-12-00361-t003:** Late diagnosis and advanced HIV disease relationship with other variables studied.

Variables	Late Diagnosis			Advanced HIV Disease		
	Patients with Late Diagnosis N = 138	Patients without Late Diagnosis N = 135	*p*-Value	Patients with Advanced HIV Disease N = 94	Patients without Advanced HIV Disease N = 179	*p*-Value
Sex N (%)			0.043			0.481
Male	113 (48.1)	122 (51.9)		79 (33.6)	156 (66.4)	
Female	25 (65.8)	13 (34.2)		15 (39.5)	23 (60.5)	
Exitus N (%)			0.682			1.000
Yes	2 (40.0)	3 (60.0)		2 (40.0)	3 (60.0)	
No	136 (50.7)	132 (49.3)		92 (34.3)	176 (65.7)	
Viral hepatitis co-infection Hepatitis B virus N (%)			0.684			0.481
Yes	4 (66.7)	2 (33.3)		3 (50.0)	3 (50.0)	
No	134 (50.2)	133 (49.8)		91 (34.1)	176 (65.9)	
Hepatitis C virus N (%)			0.069			0.032
Yes	11 (73.3)	4 (26.7)		9 (60.0)	6 (40.0)	
No	127 (49.2)	131 (50.8)		85 (32.9)	173 (67.1)	
Age (years)			<0.001			<0.001
Mean	42.15	36.91		43.09	37.70	
Median	42.47	35.70		43.08	36.58	
Viral load (log copies/mL)			<0.001			<0.001
Mean	5.36	4.82		5.54	4.86	
Median	5.30	4.78		5.50	5.54	
HIV transmission mode N (%)			0.001			<0.001
MSM	64 ^a^ (40.5)	94 ^a^ (59.5)		40 ^a^ (25.3)	118 ^a^ (74.7)	
Heterosexual	46 ^b^ (59.7)	31 ^b^ (40.3)		33 ^b^ (42.9)	44 ^b^ (57.1)	
IVDU	10 ^a,b^ (66.7)	5 ^a,b^ (33.3)		6 ^a,b^ (40.0)	9 ^a,b^ (60.0)	
Unknown	18 ^b^ (78.3)	5 ^b^ (21.7)		15 ^b^ (65.2)	8 ^b^ (34.8)	
Race N (%)			0.065			0.020
Caucasian	108 ^a^ (49.3)	111 ^a^ (50.7)		73 ^a,b^ (33.3)	146 ^a,b^ (66.7)	
Latino	28 ^a^ (62.2)	17 ^a^ (37.8)		21 ^b^ (46.7)	24 ^b^ (53.3)	
Black	2 ^a^ (22.2)	7 ^a^ (77.8)		0 ^a^ (0.0)	9 ^a^ (100.0)	
Level of diagnostic care N (%)			0.069			0.004
Primary care	65 ^a^ (48.5)	69 ^a^ (51.5)		38 ^a^ (28.4)	96 ^a^ (71.6)	
Specialised care consultation	32 ^a^ (57.1)	24 ^a^ (42.9)		23 ^a,b^ (41.1)	33 ^a,b^ (58.9)	
Inpatient	29 ^a^ (60.4)	19 ^a^ (39.6)		26 ^b^ (54.2)	22 ^b^ (45.8)	
Private consultation	8 ^a^ (32.0)	17 ^a^ (68.0)		5 ^a,b^ (20.0)	20 ^a,b^ (80.0)	
NGO/“ADOS”	3 ^a^ (75.0)	1 ^a^ (25.0)		2 ^a,b^ (50.0)	2 ^a,b^ (50.0)	
Rapid Test	1 ^a^ (16.7)	5 ^a^ (83.3)		0 ^a,b^ (0.0)	6 ^a,b^ (100.0)	

MSM: men who have sex with men; IVDU: intravenous drug users; NGO: non-governmental organization; ADOS: axencia de doazón de órganos e sangue (Galician blood and tissue donation agency). The first category has been taken as the reference modality. Superscripts a,b: for multiple comparisons, each different superscript letter indicates a subset in the category, whose proportions of late diagnosis and advanced HIV disease differ significantly from each other at the Bonferroni-corrected 0.05.

**Table 4 healthcare-12-00361-t004:** Late diagnosis, advanced HIV disease, and AIDS relationship with previous HIV serology.

Variables	Previous HIV Serology		*p*-Value
	Yes N (%)	No N (%)	
Late diagnosis	30 (21.7)	108 (78.3)	0.002
Advanced HIV disease	17 (18.1)	77 (81.9)	0.001
AIDS	18 (18.6)	79 (81.4)	0.002

AIDS: Acquired Immunodeficiency Syndrome.

**Table 5 healthcare-12-00361-t005:** Late diagnosis/advanced HIV disease adjusted logistic regression model vs. other variables studied.

Variables	Late Diagnosis			Advanced HIV Disease		
	OR	95% CI OR	Sig.	OR	95% CI OR	Sig.
Age (years)	1.032	1.006–1.058	0.017	1.030	1.003–1.059	0.029
Viral load (log copies/mL)	2.115	1.528–2.929	0.000	2.921	1.999–4.268	0.000
Hepatitis C virus	--	--	--	10.015	1.691–59.320	0.011
HIV transmission mode						
MSM	--	--	0.013	--	--	0.002
Heterosexuals	2.156	1.178–3.945	0.013	2.330	1.205–4.506	0.012
IVDU	2.198	0.680-7.100	0.188	0.295	0.042–2.075	0.220
Unknown	4.135	1.349–12.676	0.013	4.772	1.666–13.672	0.004

MSM: men who have sex with men; IVDU: intravenous drug users; OR: odds ratio; Sig.: significance.

**Table 6 healthcare-12-00361-t006:** Health contact/missed diagnostic opportunities relationship with other variables studied.

Variables	Emergency Department	Scheduled Hospitalisation	Specialised Care Consultation	Primary Care	Total
Health contact (N (%))	432 (21.7)	7 (0.4)	283 (14.2)	1265 (63.7)	1987 (100)
Health contact Media (SD)	1.58 (2.86)	0.03 (0.22)	1.04 (1.36)	4.63 (3.00)	7.28 (5.64)
Missed diagnostic opportunities (N (%))	55 (23.6)	0 (0.0)	43 (18.5)	135 (57.9)	233 (100)
Missed diagnostic opportunities by groups N (%)					
0	238 (87.2)	273 (100.0)	235 (86.1)	174 (63.7)	153 (56.0)
1–2	33 (12.1)	0 (0.0)	38 (13.9)	89 (32.6)	88 (32.2)
3–4	2 (0.7)	0 (0.0)	0 (0.0)	9 (3.3)	28 (10.3)
>4	0 (0.0)	0 (0.0)	0 (0.0)	1 (0.4)	4 (1.5)
Relationship between health contact and missed diagnostic opportunities and late diagnosis/advanced HIV disease					
Health contact			Median	IQR	*p*-value
	Late diagnosis	Yes	6.5	4–10	0.386 ^1^
		No	6	3–10	
	Advanced HIV disease	Yes	7	4–10	0.039 ^1^
		No	6	3–10	
Missed diagnostic opportunities			Mean	SD	*p*-value
	Late diagnosis	Yes	1.1	1.4	<0.001 ^1^
		No	0.6	1.0	
	Advanced HIV disease	Yes	1.2	1.5	<0.001 ^1^
		No	0.6	1.0	
Relationship between missed diagnostic opportunities and other variables studied					
Variables		Mean	Median	SD	*p*-value
Sex	Male	0.7	0.0	1.1	0.012 ^1^
	Female	1.4	1.0	1.7	
Hepatitis B virus	No	267	0.8	1.2	0.848 ^1^
	Yes	6	0.5	0.5	
Hepatitis C virus	No	0.8	0.0	1.2	0.032 ^1^
	Yes	1.5	1.0	1.5	
HIV transmission mode	MSM	0.7	0.0	1.0	0.114 ^2^
	Heterosexual	0.8	0.0	1.1	
	IVDU	1.5	1.0	1.5	
	Unknown	1.3	0.0	2.1	
Level of diagnostic care	Primary care	0.8	0.0	1.3	0.072 ^2^
	Specialised care consultation	1.0	1.0	1.1	
	Inpatient	0.9	0.0	1.4	
	Private consultation	0.4	0.0	0.9	
	NGO/“ADOS”	0.3	0.0	0.5	
	Rapid test	0.3	0.0	0.5	
Relationship between age and missed diagnostic opportunities					
Age	Spearman’s Rho		95% CI (two-tailed) ^a,b^		Sig.
	0.011		−0.112/0.133		0.861
Relationship between health contact and previous HIV serology					
Previous serology		N	Mean	Median	*p*-value
	Yes	83	8.1	7.0	0.281
	No	190	6.9	6.0	

CI: confidence interval; SD: standard deviation; IQR: interquartile range; MSM: men who have sex with men; IVDU: intravenous drug users; NGO: non-governmental organization; ADOS: axencia de doazón de órganos e sangue (Galician blood and tissue donation agency). 1: Mann–Whitney U test. 2: Kruskal–Wallis test. Spearman’s correlation: a. Estimation based on Fisher’s r to z transformation. b. The standard error estimate is based on the formula proposed by Fieller, Hartley and Pearson.

## Data Availability

Data are contained within the article.

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
