# Peer review of "Analysing Early Diagnosis Strategies for HIV Infection: A Retrospective Study of Missed Diagnostic Opportunities"

_healthcare, 2024, doi:10.3390/healthcare12030361_

Round 1

Reviewer 1 Report

Comments and Suggestions for Authors

The study, titled “Analysis of new Human Immunodeficiency Virus diagnoses and missed opportunities of screening strategies” is a retrospective investigation that assesses the relationship between Late Diagnoses, Advanced Disease, baseline characteristics, and the detection of Missed Diagnostic Opportunities. This study is a relevant to public health, as it has the potential to contribute to the optimization of screening strategies for early HIV diagnosis. However, the paper needs some modification and adjustment to make it clearer. Below are a few suggestions to make the study clear and complete.

Title:
The title does not represent what the study shows. I suggest reviewing the title. Here are a couple of suggestions: “Optimizing Early Diagnosis Strategies for HIV Infection: A Retrospective Analysis of Missed Diagnostic Opportunities, Late Diagnoses, and Advanced Disease Incidence”, “Optimizing Early Diagnosis Strategies for HIV Infection: A Retrospective Analysis of Missed Diagnostic Opportunities”

Results:

1)    Exclude the first sentence of the results “This section may be divided by subheadings. It should provide a concise and precise description of the experimental results, their interpretation, as well as the experimental conclusions that can be drawn”

2)    Ensure that all tables follow the recommended table format (side lines, columns, and line structure). Review all abbreviations in the legend and confirm that the numbers and percentages add up.

3)    Table 1: include the definition of the clinical category in the legend.

4)    Table 2 Clarify how the percentage is calculated and ensure that the numbers add up.

5)    Table 3: “Patients without EA” what’s EA? Shouldn’t it be AD?

6)    Table 4: add the meaning of “ED” in the legend. Revise the topic title “Relationship between MDO with vs other variables studied” for clarity.

Discussion

In the discussion, address some point where it’s mentioned that the results were found in other studies, but the result is not discussed. One example is Page 12, line 66, line 84 where the location or other characteristics of the studies could be important to mention, specially considering that HIV present different healthcare approaches depending on the country and risk group.

Author Response

Thank you very much for taking the time to review this manuscript. Please see the attachment.

Reviewer 2 Report

Comments and Suggestions for Authors

The manuscript "Analysis of new Human Immunodeficiency Virus diagnoses and missed opportunities of screening strategies" presents a concise but clear assessment of current strategies to identify HIV infection early as a public health practice to reduce transmission.  I enjoyed the paper and the issues it presents regarding the limitations of the current guidelines on when to offer HIV testing and the burdens placed on providers in evaluating patients needs and risks.

A couple of minor clarifications could help strengthen the paper:

1. The results section seems to contain a few opening lines (111-113) that are not part of the manuscript.

2. The many acronyms in this paper make it a challenge to always follow the argument. Could some acronyms be eliminated that are used less often, like HTX? And tables would be clearer if they didn't use acronyms.

3. What is considered Latin origin in Spain? Is it the same as Latino in the US, in other words someone with origins that trace back to central and Southern America?

4. Could you offer some details on the guidelines for offering an HIV test. You reference it, but it would be helpful to understand how much the results suggest a need to revise these guidelines versus a better education of providers to follow the guidelines. Figure 1 had a great deal of detail that didn't make much sense without having the details of the guidelines for when HIV testing should be offered when patients present with other conditions.

5. Along the same lines, could the discussion and conclusion offer more indications of what should be altered to address the identified issues. The discussion indicates how much the findings align with other studies, but doesn't present an indication for where the failure lies in the system and what might be done to address it.

 6. Finally, the link between HCV and HIV, indicating that patients with HCV diagnoses present with more advanced HIV. Could you explain why you think this is? With regards to failures of the system to catch an HIV diagnosis it seems odd, however, the relationship between the two diseases suggests that HCV may worsen the progression of HIV, such that it isn't a missed diagnosis or delayed diagnosis, but rather a biological relationship of comorbidities. Similarly, could you explain some of the other findings, like why women are more likely to have late diagnoses, or why people who inject drugs or those who have an unknown transmission route are more like to have late diagnosis?

Author Response

(The authors gave the same response as above.)

Reviewer 3 Report

Comments and Suggestions for Authors

The topics covered by the authors are truly relevant and of interest to many readers, specifically in low- and middle-income countries where the burden of HIV is the greatest.

The authors’ methodology is confusing. A cleaner approach would be to exclude people with prior serology because they are different from those who have been tested for the first time and considered as late diagnosis or advanced diagnosis. In fact, the authors should have anticipated that possibility which their results have confirmed.

The distinction between late diagnosis, AIDS and advanced HIV disease is also confusing. Late diagnosis as defined by CD4 <350 cells/mm3 include advanced disease (CD4 count <200 which is also used as an AIDS defining criteria). All these categories are not mutually exclusive and made the analysis confusing. The authors could see that risk factors for late diagnosis and advanced diseases were similar.

The four clusters by diagnostic test indicator status are not straightforward. I was wondering the reason the authors did not use the WHO clinical staging that would allow more comparisons as it is commonly used by HIV researchers.

Multivariable logistic regression was proposed but the authors did not show a table with both unadjusted and adjusted odds ratios.

Removing participants with prior serology will reduce the number of study participants and therefore worsen one of the study limitations.

I have not seen references with advanced HIV disease as keywords.

See additional inputs as highlights and comments.

Comments on the Quality of English Language

The quality of the English language should be improved.

Author Response

(The authors gave the same response as above.)
